Development and validation of a multi-parameter nomogram for predicting prostate cancer: a retrospective analysis from Handan Central Hospital in China

Nan Libin 1 nanlibin@126.com
Guo Kai 2
Li Mingmin 3
Wu Qi 1
Huo Shaojun 1
1 Department of Urology, Handan Central Hospital , Handan, Hebei , China
2 Cardiac Department, Turku City Hospital , Turku, Varsinais-suomi , Finland
3 Out-patient Department, Handan Central Hospital , Handan, Hebei , China
Fionda Bruno
Electronic publication date: 2022 Mar 2
Publication date: 2022
Volume: 10
Electronic Location ID: e12912
Received 2021 Nov 29; Accepted 2022 Jan 19
Copyright: © 2022 Nan et al.
Copyright year: 2022
Copyright holder: Nan et al.
License: This is an open access article distributed under the terms of the Creative Commons Attribution License, which permits unrestricted use, distribution, reproduction and adaptation in any medium and for any purpose provided that it is properly attributed. For attribution, the original author(s), title, publication source (PeerJ) and either DOI or URL of the article must be cited.
License URL: https://creativecommons.org/licenses/by/4.0/

Keywords: Prostatic neoplasms, Prostate-specific antigen, Nomograms

Funding: The authors received no funding for this work.

==============================
Background

To explore the possible predicting factors related to prostate cancer and develop a validated nomogram for predicting the probability of patients with prostate cancer.

Method

Clinical data of 697 patients who underwent prostate biopsy in Handan Central Hospital from January 2014 to January 2020 were retrospectively collected. Cases were randomized into two groups: 80% (548 cases) as the development group, and 20% (149 cases) as the validation group. Univariate and multivariate logistic regression analyses were performed to determine the independent risk factors for prostate cancer. The nomogram prediction model was generated using the finalized independent risk factors. Decision curve analysis (DCA) and the area under receiver operating characteristics curve (ROC) of both development group and validation group were calculated and compared to validate the accuracy and efficiency of the nomogram prediction model. Clinical utility curve (CUC) helped to decide the desired cut-off value for the prediction model. The established nomogram with Prostate Cancer Prevention Trial Derived Cancer Risk Calculator (PCPT-CRC) and other domestic prediction models using the entire study population were compared.

Results

The independent risk factors determined through univariate and multivariate logistic regression analyses were: age, tPSA, fPSA, PV, DRE, TRUS and BMI. Nomogram prediction model was developed with the cut-off value of 0.31. The AUC of development group and validation group were 0.856 and 0.797 respectively. DCA exhibits consistent observations with the findings. Through validating our prediction model as well as other three domestic prediction models based on the entire study population of 697 cases, our prediction model demonstrated significantly higher predictive value than all the other models.

Conclusion

The nomogram for predicting prostate cancer can facilitate more accurate evaluation of the probability of having prostate cancer, and provide better ground for prostate biopsy.

Introduction

With the prolongation of life span, the change of dietary structure and other factors, the incidence rate of prostate cancer has increased over the years (Siegel et al., 2021), making it more important to emphasize early screening for prostate cancer. In clinical practice, prostate biopsy is currently regarded as the gold standard for the diagnosis of prostate cancer. Whether to perform prostate puncture commonly depends on the level of serum prostate specific antigen (PSA) and its related parameters, digital rectal examination (DRE) and imaging results. However, the current primary non-invasive prostate cancer screening methods have resulted in an unsatisfactory amount of unnecessary prostate biopsy cases. Given the nature of being invasive and relatively expensive, prostate biopsy needs better ground to promote quality of life for patients. Many of the domestic pioneering scholars have proposed prostate cancer prediction models based on multiple clinical parameters in recent years, such as Tang et al. (2013) (model 1), Huang et al. (2014) (model 2), Li et al. (2016) (model 3). There is a significant difference as for the morbidity rate of prostate cancer when we talk about different races or geographical distribution (Siegel et al., 2021). Therefore, prediction models which have been established from other regions such as PCPT-CRC (Thompson et al., 2006) might not be fit for our study population. Meanwhile, the types of variables utilized for prediction model still need to be further explored, and the existing prediction models require more regional data analysis and validation. Based on the single center data source of Handan Central Hospital, this study aims to build a nomogram prediction model for prostate cancer, and promote early detection for patients with possible prostate cancer.

Materials and Methods

Study population

In this study, data were retrospectively collected from a total of 789 patients who underwent transrectal ultrasound (TRUS)-guided transrectal prostate puncture (12+X core) in Handan Central Hospital from January 2014 to January 2020. All prostate biopsies were performed by an experienced urologist from our department who has practiced his profession for more than 10 years. Pathological specimens were initially processed by 2 pathologists from the pathology department. Final pathological reports were provided by the senior pathologist after reviewing and the initial reports. Data screening was performed to exclude the following cases: First, 18 patients were previously diagnosed with prostate cancer or being surgically treated. Second, 35 patients were taking 5-alpha-reductase inhibitor/drug for treating endocrine dyscrasia in prostate cancer. Third, 15 patients had results of tPSA >100 ng/mL. Fourth, 24 cases had insufficient data. Eventually 697 cases were included. The median age was 71 years (40–95 years), and the median tPSA is 13.6 ng/mL (0.2–100 ng/mL). To develop the nomogram, 548 (80%) cases were randomly selected and rest of 149 (20%) cases are set as the validation group by SPSS software.

Ethics statement

The study was approved by ethics committee of Handan Central Hospital. Consent was obtained from all patients prior to transrectal ultrasound-guided prostate puncture.

Methods

Baseline data were retrospectively collected from the clinical digital information system including age, tPSA, fPSA, DRE results, TRUS findings, history of hypertension and diabetes. Prostate puncture was performed with one or more of the following criteria (Sun, 2016): (1) tPSA > 10 ng/ml; (2) %fPSA < 0.16 when tPSA is within the range of 4–10 ng/ml; (3) DRE and/or TRUS showed abnormal findings. PV was calculated by the formula: PV (ml) = anterior/posterior diameter (cm) × left/right diameter (cm) × upper/lower diameter (cm) × 0.52. PSAD value was further calculated by dividing tPSA by PV. As an extended PSA-related figure, %fPSA was calculated as well (fPSA/tPSA). BMI figures of all cases were calculated through weight (kg) divided by the square of height (m2), and further sorted into four intervals according to the World Health Organization (WHO) guidelines for Asian population (Feng et al., 2019): 56 underweight (<18.5 kg/m2) cases, 255 normal weight (18.5–22.9 kg/m2) cases, 266 overweight (23.0–27.4 kg/m2) cases, and 120 obese (≥27.5 kg/m2) cases. Since there are only 56 underweight cases, in this study they were combined with normal weight group.

The three domestic prediction model equations are listed as follows: Model 1 (Tang et al., 2013): e−1.163+0.033Age+1.032DRE−2.821LogPV+2.292LogPSA1+e−1.163+0.033Age+1.032DRE−2.821LogPV+2.292LogPSA

Model 2 (Huang et al., 2014): e−3.577+0.054(Age)−3.714(f/tPSA)−1.324(Ln(PV))+0.977(Ln(PSA))+1.698(DREfindings)+0.458(hypoechoic)1+e−3.577+0.054(Age)−3.714(f/tPSA)−1.324(Ln(PV))+0.977(Ln(PSA))+1.698(DREfindings)+0.458(hypoechoic)

Model 3 (Li et al., 2016): e−5.348+0.09(Age)−0.043(tPSA)+0.439(fPSA)−0.015(PV)−8.718(f/tPSA)+2.614(PSAD)1+e−5.348+0.09(Age)−0.043(tPSA)+0.439(fPSA)−0.015(PV)−8.718(f/tPSA)+2.614(PSAD)

Statistical analysis

The Kolmogorov–Smirov test was used to assess the normal distribution of variables. For comparison purpose, T-test and U Mann–Whitney test were used to evaluate parametric and nonparametric continuous variables, respectively. Chi-squared test was used to compare qualitative variables. Univariate logistic regression analysis was utilized to determine the independent risk factors. Multicollinearity diagnostics were conducted to determine whether the independent factors have no multicollinearity (Variance Inflation Factor <10). The remaining independent risk factors were included in the multivariate logistic regression analysis, and the forward stepwise method was used to further determine the variables for developing the model. The prediction model was graphically presented as a nomogram for clinical use. Clinical utility curve was used to demonstrate the number of PCa cases which were left undiagnosed and the number of non-PCa cases which were spared from unnecessary prostate biopsies. Then Youden index was used to determine the desired cut-off value. Diagnostic accuracy was quantified as the area under the ROC curve (AUC). Statistical differences between AUCs were compared using the DeLong method. Based on the optimal cut-off value, the diagnostic accuracy was assessed using sensitivity, specificity, positive predictive value (PPV) and negative predictive value (NPV). The calibration curve was utilized to assess the agreement of the nomogram-predicted probability with the actual observed probability. Decision curve analysis (DCA) was conducted to determine the net benefit derived from the use of the created nomogram. SPSS 22.0, medcalc19.0.7 and R language 4.0.0 statistical software were used to process the data. A P-value < 0.05 is considered statistically significant.

Results

Patient demographics

Out of a total amount of 697 cases, 504 (72.3%) cases were non-prostate cancer patients, 193 (27.7%) cases were found with prostate cancer according to the biopsy results. Specifically in non-prostate cancer group, there are 454 cases of benign prostatic hyperplasia, 36 cases of prostatic intraepithelial neoplasia, and 14 cases of prostatitis. Comparison of the baseline data indicated that there is no significant difference between the development and validation group in the general situation of patients, characters of age, tPSA, fPSA, and other indicators (P > 0.05) (Table 1). In normal weight, overweight and obese BMI intervals, the rate of having prostate cancer is 22.5% (70/311), 29.3% (78/266), and 37.5% (45/120) respectively, indicating a statistical difference among the three groups (χ2 = 10.293, P < 0.05).

Table 1 Baseline clinical characteristics of the development group and validation group.

Parameter	Total	Development group	Validation group	Z/χ2	P-value	
Number of patients	697	548	149			
Age (year)	71 (66∼77)	71 (66∼77)	72 (65∼78)	−0.751	0.453	
tPSA (ng/ml)	13.6 (5.6∼30.7)	13.6 (5.6∼30.5)	14.6 (5.4∼32.8)	−0.279	0.78	
fPSA (ng/ml)	1.8 (0.8∼4.3)	1.8 (0.73∼4.2)	2.0 (0.95∼4.5)	−1.071	0.284	
PV (ml)	47.9 (36.0∼69.2)	46.6 (35.6∼68.2)	50.4 (38.1∼72.4)	−1.556	0.12	
%fPSA	0.14 (0.11∼0.18)	0.13 (0.1∼0.17)	0.15 (0.12∼0.20)	−3.088	0.002	
PSAD	0.26 (0.1∼0.60)	0.25 (0.11∼0.61)	0.26 (0.09∼0.55)	−0.377	0.706	
DRE [n(%)]				0.046	0.83	
Normal	557	437 (80)	120 (81)			
Suspect cancer	140	111 (20)	29 (19)			
TRUS finding * [n(%)]				0.47	0.493	
Negative	475 (68)	370 (68)	105 (70)			
Positive	222 (32)	178 (32)	44 (30)			
BMI(kg/m2) [n(%)]				1.959	0.376	
≤22.9	311 (45)	237 (43)	74 (50)			
23.0∼27.4	266 (38)	214 (39)	52 (35)			
≥27.5	120 (17)	97 (18)	23 (15)			
Hypertension [n(%)]				0.803	0.37	
No	322 (46)	258 (47)	64 (43)			
Yes	375 (54)	290 (53)	85 (57)			
Diabetes [n(%)]				0.256	0.613	
No	575 (82)	450 (82)	125 (84)			
Yes	122 (18)	98 (18)	24 (16)			
Note:

tPSA, total prostate-specific antigen; fPSA, free prostate-specific antigen; PV, prostate volume; DRE, digital rectal examination; TRUS, transrectal ultrasound; *Low-echogenicity in the peripheral zone of the prostate was defined as ‘positive’; other findings were defined as ‘negative’.

Nomogram development

Univariate logistic regression analysis of the development group showed that age, tPSA, fPSA, PV, DRE, TRUS, BMI were statistically significant (P < 0.05), whereas hypertension and diabetes were not significantly related to prostate cancer. Multicollinearity diagnostics were performed utilizing the abovementioned independent risk factors, and the VIFs were 1.042, 2.338, 2.405, 1.121, 1.061, 1.043, 1.022 respectively, suggesting that there was no multicollinearity among the seven independent risk factors. Lastly, multivariate logistic regression confirmed that age, tPSA, fPSA, PV, DRE, TRUS, and BMI served as independent risk factors for prostate cancer (P < 0.05) which could be included in the development of predicting model (Table 2).

Table 2 Univariate and multivariate logistic regression models in the development group.

Variable	Univariate analysis	Multivariate analysis	
OR (95% CI)	P-value	OR (95% CI)	P-value	
Age	1.036 [1.010∼1.063]	0.007	1.039 [1.005∼1.074]	0.024	
tPSA	1.040 [1.030∼1.050]	0.01	1.026 [1.010∼1.042]	0.002	
fPSA	1.256 [1.187∼1.233]	<0.001	1.215 [1.106∼1.336]	<0.001	
PV	0.987 [0.979∼0.995]	0.01	0.972 [0.961∼0.983]	<0.001	
DRE	3.131 [2.024∼4.843]	<0.001	3.185 [1.798∼5.641]	<0.001	
TRUS	4.1 [2.754∼6.104]	<0.001	4.560 [2.773∼7.500]	<0.001	
BMI	1.466 [1.137∼1.889]	0.003	1.852 [1.337∼2.567]	<0.001	
Hypertension	0.988 [0.677∼1.442]	0.951			
Diabetes	0.853 [0.515∼1.412]	0.536			
Note:

tPSA, total prostate-specific antigen; fPSA, free prostate-specific antigen; PV, prostate volume; DRE, digital rectal examination; TRUS, transrectal ultrasound; OR, odds radio; CI, confidence interval.

Based on the multivariate logistic regression analysis, the predicting equation was generated with the calculated coefficients: logit(1/1−P) = −4.934 + 0.038 × Age + 0.025 × tPSA + 0.195 × fPSA-0.028 × PV + 1.158 × DRE + 1.517 × TRUS + 0.617 × BMI (Table 3). According to the variables and their corresponding regression coefficients, the nomogram representing the predicting model is then established (Fig. 1).

Figure 1 Nomogram for prostate cancer prediction model.

Table 3 Result of multivariate logistic regression analysis in the development group.

Variable	Coefficient	SE	Wald	OR	95% CI	P-value	
Age	0.038	0.017	5.069	1.039	[1.005∼1.074]	0.024	
tPSA	0.025	0.008	9.853	1.026	[1.010∼1.042]	0.002	
fPSA	0.195	0.048	16.28	1.215	[1.106∼1.336]	<0.001	
PV	−0.028	0.006	22.599	0.972	[0.961∼0.983]	<0.001	
DRE	1.158	0.292	15.782	3.185	[1.798∼5.641]	<0.001	
TRUS	1.517	0.254	35.734	4.56	[2.773∼7.500]	<0.001	
BMI	0.617	0.166	13.714	1.852	[1.337∼2.567]	<0.001	
Constant	−4.934	1.312	14.15	0.007		<0.001	
Note:

tPSA, total prostate-specific antigen; fPSA, free prostate-specific antigen; PV, prostate volume; DRE, digital rectal examination; TRUS, transrectal ultrasound; SE, Standard error; OR, odds radio; CI, confidence interval.

To properly use the nomogram, points are collected from each category (age, tPSA, fPSA, PV, DRE, TRUS and BMI) according to an individual’s corresponding test results, and the total points indicate a diagnostic possibility of having prostate cancer.

Nomogram validation

The clinically commonly used parameters tPSA, f/tPSA, and PSAD are used to determine the diagnostic value of the prediction model through comparing their ROC curves. The AUC values of development group, tPSA, %fPSA, PSAD were 0.856, 0.713, 0.624, 0.761 respectively (Fig. 2 and Table 3), and the AUC values of validation group, tPSA, %fPSA and PSAD were 0.797, 0.662, 0.624, 0.673 respectively (Fig. 3 and Table 4). Predicting model indicated statistically significant better diagnostic value against other risk factors in both development group and validation group (P < 0.05). Clinical utility curve (CUC) demonstrated the dynamics between the percentage of cases left undiagnosed and percentage of cases saved from unnecessary biopsies at any threshold of probabilities (Figs. 4 and 5). The cut-off value for the prediction model was selected as 0.31 when Youden index reached maximum value. Sensitivity, specificity, positive predictive value, negative predictive value, false negative rate and false positive rate were 73%, 85.8%, 65.5%, 89.6%, 27% and 14.2% respectively. Applying the cut-off value 0.31 into the validation group resulted the sensitivity, specificity, positive predictive value, negative predictive value, false negative rate and false positive rate, which were 62.2%, 87.5%,68.3%, 84.3%, 37.8% and 12.5% respectively (Table 4).

Figure 2 ROC curve presenting the discrimination power of the nomogram (development group).

Figure 3 ROC curve presenting the discrimination power of the nomogram (validation group).

Figure 4 Clinical utility curve of the development group.

Figure 5 Clinical utility curve of the validation group.

Table 4 Diagnostic values of model and clinical parameters in the development group and validation group for the results of prostate biopsy.

Variable	Cutoff	Youden index	SEN	SPE	PPV	NPV	FNR	FPR	AUC	95% CI	P-value	
(%)	(%)	(%)	(%)	(%)	(%)	(AUC)	
Model (dev)	>0.31	0.59	73	85.8	65.5	89.6	27	14.2	0.856	[0.824∼0.885]	<0.001	
tPSA	>31.5	0.36	49.3	86.5	57.5	82.2	50.7	13.5	0.713	[0.673∼0.751]	<0.001	
%fPSA	<0.17	0.28	46.6	81.5	48.3	80.5	53.4	18.5	0.624	[0.582∼0.665]	<0.001	
PSAD	>0.44	0.46	65.5	80	54.8	86.3	34.5	20	0.761	[0.723∼0.797]	<0.001	
Model (val)	>0.31	0.49	62.2	87.5	68.3	84.3	37.8	12.5	0.797	[0.724∼0.859]	<0.001	
tPSA	>31.5	0.24	44.4	80	48.8	76.9	55.6	20	0.662	[0.580∼0.737]	<0.001	
%fPSA	<0.17	0.26	53.3	73.1	46.2	78.4	46.7	26.9	0.624	[0.541∼0.702]	<0.001	
PSAD	>0.44	0.31	53.3	77.9	51.1	79.4	46.7	22.1	0.673	[0.592∼0.748]	<0.001	
Note:

dev, development group; val, validation group; tPSA, total prostate-specific antigen; %fPSA, the ratio of fPSA to tPSA; PSAD, prostate-specific antigen density; SEN, sensitivity; SPE, specificity; PPV, positive predictive value; NPV, negative predictive value; FNR, false negative rate; FPR, false positive rate; AUC, area under the curve; CI, confidence interval.

The calibration plot demonstrated an outstanding correlation between the predicted and actual probability in both development group and validation group, in which the predicted probability and actual probability lines are closely aligned with remarkable P-values of 0.374 in development group and 0.236 in validation group. The intercept and slope in the development group were −0.057 and 1.044, while in the validation group were 0.178 and 0.877 (Figs. 6 and 7).

Figure 6 Calibration curve of the prediction model in the development group.

Figure 7 Calibration curve of the prediction model in the validation group.

When the threshold range is within 11–69% and 75–81% in the validation group, and 4–83% in the development group, the net benefit of our prediction model is higher than that of the other clinically commonly used parameters (Figs. 8 and 9). The cut-off value of 0.31 which resulted from ROC curve analysis was utilized as the determined threshold probability. The net benefit and net reduction of our prediction model are both higher than that of the other three diagnostic parameters in both development and validation groups (Table 5).

Figure 8 Decision curve analysis of the prediction model and other variables in the development group.

predmodelA: prediction model; predmodelB: tPSA; predmodelC: %fPSA; predmodelD: PSAD.

Figure 9 Decision curve analysis of the prediction model and other variables in the validation group.

predmodelA: prediction model; predmodelB: tPSA; predmodelC: %fPSA; predmodelD: PSAD.

Table 5 Net benefit and reduction of the prediction model and other variables in the development and validation group.

Threshold	dev	31	val	31	
Probability (%)	
Net benefit (%)	Model	15.1	Model	13.9	
tPSA	7.9	tPSA	6.5	
%fPSA	4.6	%fPSA	5.5	
PSAD	10.3	PSAD	8.6	
Treat all	−5.8	Treat all	−1.2	
Net reduction (%)	Model	46.5	Model	33.5	
tPSA	30.5	tPSA	17	
%fPSA	28.5	%fPSA	14.9	
PSAD	35.8	PSAD	21.7	
Note:

dev, development group; val, validation group.

Together with the other three domestic prediction models, our developed prediction model is further validated through utilizing the validation group of 149 cases. The AUC of our prediction model, domestic model 1, domestic model 2 and domestic model 3 are 0.797, 0.739, 0.753 and 0.694, respectively. Separately, enable to compare with PCPT-CRC, the validation group was modified according to the limitations for entries of PCPT-CRC. Specifically, 1 case was exempted for being over 55 years old, 12 cases were ruled out for having tPSA more than 50 ng/mL, resulting a separate validation group with 136 cases. The AUC of our prediction model and PCPT-CRC using such new validation group were 0.793 and 0.668 (Table 6). Statistically significant differences were found comparing our prediction model with PCPT-CRC and domestic prediction model 3 (P < 0.05). However, it was not significantly different compared with domestic model 1 and 2 (Table 7).

Table 6 Diagnostic accuracy of our model and other models using validation group.

Prediction model	AUC	95% CI	P-value (AUC)	
Our model (val)	0.797	[0.724∼0.859]	<0.001	
Domestic Model 1	0.739	[0.661∼0.808]	<0.001	
Domestic Model 2	0.753	[0.676∼0.820]	<0.001	
Domestic Model 3	0.694	[0.613∼0.766]	<0.001	
Our model (val*)	0.793	[0.715∼0.857]	<0.001	
PCPT model	0.668	[0.582∼0.746]	<0.001	
Note:

AUC, area under the curve; CI, confidence interval; val, validation group (149 cases); val*, validation group (136 cases).

Table 7 Comparison of diagnostic values of other prediction models with that of our model.

Prediction model	AUC	P-value	
Our model (val)	0.797	N/A	
Domestic Model 1	0.739	0.1471	
Domestic Model 2	0.753	0.2424	
Domestic Model 3	0.694	0.0148	
Our model (val*)	0.793	N/A	
PCPT-CRC	0.668	0.032	
Note:

AUC, area under the curve; val, validation group (149 cases); val*, validation group (136 cases).

Discussion

Prostate biopsy is the gold standard for diagnosing prostate cancer, but there are risks of complications such as hematuria, bloody stool, urinary retention and infection. Therefore, various international researchers intended to reduce prostate puncture rate through utilizing different types of variables to develop predicting models and presenting them as forms of classification tree model (Eifler et al., 2013), artificial neural network model (Cai & Jiang, 2014) and nomogram model for better comprehension and feasibility. Among the presenting models, nomogram model has the characteristics of integrating multiple predictive variables, quantifying the contribution of related risk factors, and presenting the predicting model in geometric form for better accessibility and utility. Even though many facilities have established their predicting models such as Prostate Cancer Prevention Trial Derived Cancer Risk Calculator (PCPT-CRC) (Thompson et al., 2006) and the Montreal (Canada) prediction model (Karakiewicz et al., 2005), it is not directly suitable for prostate cancer screening in Chinese population due to demographic differences (Wu et al., 2016). Meanwhile, people of same race living in different regions could lead to a difference in the incidence rate of prostate cancer (Gu et al., 2019). Our statistical analysis of comparing with PCPT-CRC further confirmed this observation. Therefore, it is of great significance to establish a prediction model for prostate cancer based on the population from a specific region.

Several domestic prediction models based on PSA and its related clinical parameters were established in China. Tang et al. (2013) established a prediction model integrating age, PSA, PV, and DRE with a total population of 535 cases, and the AUC was 0.848. Another domestic prediction model which developed by Huang et al. (2014) included age, PSA, PV, %fPSA, TRUS and DRE as independent risk factors, acquired the AUC of 0.853 based on a total population of 1104 cases. Lastly, with the population of 958 cases, Li et al. (2016) established a prediction model based on age, tPSA, fPSA, PV, %fPSA and PSAD, resulting the AUC of 0.854. In this study, univariate and multivariate logistic regression analysis showed that age, TPSA, fPSA, PV, DRE, TRUS and BMI were independent predictors of prostate cancer in the development group. Based on these clinical variables, a nomogram prediction model was established. Considering the progression of prostate cancer when false negative results happen, vs the complications which prostate puncture possibly brings, we have decided to take the cut-off value of AUC when the Youden index was at its optimal level. Such cut-off value of AUC in the development group was 0.31. The sensitivity, specificity, positive predictive value, negative predictive value, false negative rate, false positive rate and AUC value of development group were 73%, 85.8%, 65.5%, 89.6%, 27%, 14.2% and 0.856, respectively. By applying the cut-off value 0.31 into the validation group resulted sensitivity, specificity, positive predictive value, negative predictive value, false negative rate, false positive rate and AUC value were calculated as 62.2%, 87.5%, 68.3%, 84.3%, 37.8%, 12.5% and 0.797, respectively, which were significantly higher than PSA related parameters. By integrating the threshold of 0.31 in the development group, our prediction model saved 85.8% of unnecessary prostate biopsies with having 27% of missed positive cases. Therefore, prostate biopsy is recommended when the prediction probability is greater than 0.31, otherwise active monitoring is preferred. After the analyses of calibration curve and decision curve, the development group and validation group showed that the prediction model had high predictive ability and clinical practicability, from which decision curve analysis indicated that utilizing our prediction model to determine the necessity of conducting prostate biopsy could significantly increase the net benefit while decreasing the rate of unnecessary prostate biopsies compared with other currently commonly used clinical diagnostic parameters. Compared with other three domestic prediction models, our prediction model demonstrated a significantly higher predictive value through validating over the entire study population. This finding suggests that: (1) even with the same race, living in different regions might lead to some distinctively different profiles of prostate cancer, making it difficult for a prediction model which was developed in a specific region to maintain its predictive value in other regions, even towards people with the same race; (2) Inclusion criteria were not always identical among different prediction models; (3) different types and amount of independent risk factors included in the development of nomogram prediction model might eventually alter the predictive value. The clinical variables included in the nomogram of this study are currently commonly used in prostate screening routines. Integrating and analyzing them in a wholistic manner can further improve the accuracy of prostate cancer detection and reduce unnecessary prostate puncture without additional medical costs.

The relationship between obesity and prostate cancer is not clear. Some researchers believe that obesity is a risk factor for prostate cancer (Cao & Giovannucci, 2016), and it can increase the mortality rate of prostate cancer patients (Dickerman et al., 2017). Among different BMI intervals, statistical analysis showed that the detection rate of prostate cancer increased with the increasement of BMI level, suggesting that obesity is likely to increase the risk of prostate cancer. The possible reasons for this phenomenon are: (1) obesity can lead to chronic inflammation, which may be related to a variety of cancers, including prostate cancer (Thapa & Ghosh, 2015). (2) Obesity can lead to high levels of insulin and insulin-like receptor factor-1, which are closely related to the development of tumor (Xue et al., 2012). (3) Leptin is mainly secreted by adipose tissue (Garcia-Galiano, Borges & Allen, 2019), and the increasement of leptin level is related to tumor invasion (Candelaria et al., 2017). (4) BMI is negatively correlated with PSA (Aref et al., 2018), resulting in delayed detection of prostate cancer in obese people. After regression analysis, BMI was included as an independent predictor in the nomogram, which further confirmed that there was a correlation between obesity and prostate cancer.

In order to improve the accuracy of prostate cancer prediction model, researchers have tried to incorporate more clinical variables into their prediction models. Zhu et al. (2015) established a nomogram model for predicting prostate cancer based on Prostate Health Index (PHI), and some other researchers integrated prostate imaging reporting and data system version 2 (PI-RADSv2) into their prediction model (Niu et al., 2017). These innovative models showed great predictive values. However, due to the uneven development level of medical care in China, the detection of p2PSA has not been widely put into practice. Besides, image grading is a subjective evaluation method, and there may be different judgements from different experts. By comparing the prediction accuracy of four models from different origins, Wang et al. (2016) found that the prediction accuracy of Huang et al. (2014) had no difference with that of PCPT-CRC model (Thompson et al., 2006) and Montreal model (Karakiewicz et al., 2005), but whether there was any consistency between different races and different regions was worth further discussion. Compared with the abovementioned models, the clinical variables included in this nomogram prediction model do not need complex calculation or additional equipment, making it possible to benefit both outpatients and inpatients. At the same time, we encourage more imaging results as independent variables to be included in the nomogram prediction model and multi-center large sample cross validation of different races and regions, so as to further improve the accuracy of the nomogram prediction model.

Shortcomings: (1) this is a retrospective study, hence there may lead to unavoidable selection bias; (2) the prediction model lacks external validation; (3) the sample size of single-center study is relatively small, and further multi-center joint study is needed to validate on a larger scale.

Supplemental Information

Supplemental Information 1 Raw data of the development group.

Click here for additional data file.

Supplemental Information 2 Raw data of the validation group.

Click here for additional data file.

Supplemental Information 3 Codebook.

Click here for additional data file.

Additional Information and Declarations

Competing Interests

Author Contributions

Data Availability

The authors declare that they have no competing interests.

Libin Nan conceived and designed the experiments, performed the experiments, analyzed the data, prepared figures and/or tables, and approved the final draft.

Kai Guo conceived and designed the experiments, analyzed the data, authored or reviewed drafts of the paper, and approved the final draft.

Mingmin Li analyzed the data, authored or reviewed drafts of the paper, and approved the final draft.

Qi Wu performed the experiments, authored or reviewed drafts of the paper, and approved the final draft.

Shaojun Huo performed the experiments, authored or reviewed drafts of the paper, and approved the final draft.

The following information was supplied regarding data availability:

The raw data is available in the Supplemental File.

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
