# Peer review of "Development and validation of a multi-parameter nomogram for predicting prostate cancer: a retrospective analysis from Handan Central Hospital in China"

_PeerJ, doi:10.7717/peerj.12912_

## Round 0.1 · original submission · Major Revisions

I warmly recommend that you seriously consider the reviewers’ suggestions before resubmitting the revised version.

Reviewer 1 ·

Basic reporting

The manuscript proposes a new nomogram to predict prostate cancer from clinical parameters. The manuscript is well written and quite clear. Also, the methodology used to build the nomogram is adequate.

I have some minor concerns:

1) Author compares their nomogram only with local models, the reason is that international models did not fit well in their population, but no analysis of this was done. To be sure of that, it is necessary to validate PCPT in their population.
2) Author declares that the calibration of the model is good, I agree with them, but the basic properties of the calibration, the intercept and the slope must be comment.
3) The comparison of the nomogram with local models was performed using the entire cohort, but it must be done using separately the development and the validation cohort, otherwise, the model developed in the population is being favored.
4) Authors used the youden index to choose a cutoff point to classify prostate cancer. This choice treats false positives and false negatives equally. It is necessary to go further, it is important to know for different cut-off points how many patients with prostate cancer are not going to be correctly classified and how many biospies could be saved. You can use the clinical utility curve for this purpose. (see Borque-Fernando Á, Esteban LM, Celma A, Roche S, Planas J, Regis L, de Torres I, Semidey ME, Trilla E, Morote J. How to implement magnetic resonance imaging before prostate biopsy in clinical practice: nomograms for saving biopsies. World J Urol. 2020 Jun;38(6):1481-1491. doi: 10.1007/s00345-019-02946-w. )

Experimental design

The study design was adequate, it would only be necessary to clarify whether or not some type of matching was carried out using a propensity matching score or similar for the choice of the validation cohort.

Validity of the findings

The use of the nomogram is limited to a local population and only uses clinical variables, but the authors declare that this is its purpose and it may have a clear utility in their healthcare setting.

Additional comments

No comments

Reviewer 2 ·

Basic reporting

no comment

Experimental design

no comment

Validity of the findings

no comment

Additional comments

General comments
Dr Libin Nan et al reported that development and validation of a multi-parameter nomogram for predicting prostate cancer: A retrospective analysis.
Their nomogram is expected to provide benefits in making decisions to prostate biopsy in Hebei, China.

Major points

1. Please consider adding the part “ nomogram from one institution in Hebei China” to the title because the title is very simple to misunderstanding.
2. In the Material and Methods, about accuracy of prostate biopsy, please describe number of doctors who performed prostate biopsy and whether they were urologists.
3. In the Material and Methods, please explain and describe prostate biopsy protocol (transrectal or transperineal, number of target biopsy etc).
4. In the Material and Methods, please explain and describe accuracy of the pathological diagnosis more detail. (Number of pathologists, central pathologist or not, uropathologist or not etc).
5. In the Material and Methods, generally BMI is not considered as parameter of multiparametric risk model for detecting prostate cancer. Therefore, please consider the test of multicollinearity of risk factors including BMI.
6. In the Resutls, please explain and describe why there was a statistically significant difference only in %fPSA even though it was randomly divided into a developing group and a validation group.
7. In the Discussion, in lines 202-204, Cut-off value P=0.31 is the ‘statistically’ best detection probability for the accuracy of the model. However, it cannot be ‘clinically’ interpreted as recommending a biopsy when the P value is 0.31 or higher and monitoring below that.

Minor points
1. In the Material and Methods, In line 71 and 72, median is considered to be a more appropriate statistical expression than mean.
2. The notation of statistical P value should be expressed in italics.

---

## Round 0.2 · accepted · Accept

After carefully checking your corrections the paper is ready to be published.

Reviewer 1 ·

Basic reporting

I appreciate the effort that authors made to improve the manuscript, all my concerns have been amended

Experimental design

no comment

Validity of the findings

no comment

Reviewer 2 ·

Basic reporting

no comment

Experimental design

no comment

Validity of the findings

no comment

Additional comments

The authors have adequately responded to the comments of the reviewers and improved the manuscript. Therefore, I recommend publication. Lastly please check and correct the following contents.

1. In the Material and Methods please list the citations for the biopsy protocol in order to use this nomogram in China and maintain the similar rate of cancer detection. Because it is necessary to adopt the same biopsy method.
2. In the Material and Methods, in line107 please correct the spelling of “Smirov”. The correct name is “Smirnov”.
3. In line135, 138 and Table1-3, The notation of statistical P value should be expressed in italics.
4. Please number the figures according to the order of the annotations on the figures in the manuscript. Aren’t the numbers of figures of CUC 3 and 4 instead of 8 and 9?